# Titin-Truncating variants predispose to dilated cardiomyopathy in populations genetically similar to african and european reference populations

John DePaolo[1], Marc R. Bornstein[2], Renae Judy[1], Sarah Abramowitz[1], Shefali S. Verma[3], Michael G. Levin[2,4], Penn Medicine Biobank[¶], Zoltan Arany[2‡], Scott M. Damrauer[1,2,4,5‡*]

1 Department of Surgery, Perelman School of Medicine, University of Pennsylvania, Philadelphia, Pennsylvania, United States of America, 2 Cardiovascular Institute, Department of Medicine, Perelman School of Medicine, University of Pennsylvania, Philadelphia, Pennsylvania, United States of America, 3 Department of Pathology and Laboratory Medicine, Perelman School of Medicine, University of Pennsylvania, Philadelphia, Pennsylvania, United States of America, 4 Corporal Michael J. Crescenz VA Medical Center, Philadelphia, Pennsylvania, United States of America, 5 Department of Genetics, Perelman School of Medicine, University of Pennsylvania, Philadelphia, Pennsylvania, United States of America

¶ Membership of the Penn Medicine Biobank is provided in the Acknowledgements and S1 Text
‡ These authors are joint senior authors on this work.
* scott.damrauer@pennmedicine.upenn.edu

## Abstract

The effect of high percentage spliced in (hiPSI) *TTN* truncating variants (TTNtvs) on risk of dilated cardiomyopathy (DCM) has historically been studied among population subgroups defined by genetic similarity to European reference populations. This has raised questions about the effect of TTNtvs in diverse populations, especially among individuals genetically similar to African reference populations. To determine the effect of TTNtvs on cardiovascular disease risk, we leveraged whole exome sequencing and electronic health record data from 43,731 Penn Medicine Biobank (PMBB) participants recruited from across the Penn Medicine healthcare system. Fraction of genetic similarity to the 1000 Genomes Project (1000G) African (AFR) reference population was determined using ADMIXTURE analysis. Logistic regression was performed to evaluate the association of hiPSI TTNtvs with prevalent DCM and atrial fibrillation (Afib), and linear regression was used to evaluate the association with reduced left ventricular ejection fraction (LVEF) either using dichotomized genetically similar population subgroup analysis or integrating ADMIXTURE population fraction. When individuals were assigned to population subgroups based on genetic similarity to the 1000G reference populations, hiPSI TTNtvs conferred significant risk of DCM among those genetically similar to the 1000G European (EUR) reference population (OR=6.12, 95% confidence intervals [CI] 4.33 to 8.65, *P* < 0.001) and individuals genetically similar to the AFR reference population (OR=3.44, 95% CI 1.97 to 5.99,

**Data availability statement:** Individual-level data are not publicly available due to research participant privacy concerns; requests from accredited researchers for access to individual-level data relevant to this manuscript can be made by contacting biobank@upenn.edu

**Funding:** J.D. is supported by the American Heart Association (23POST1011251, www.heart.org); this award includes salary support. M.G.L. received support from the Institute for Translational Medicine and Therapeutics of the Perelman School of Medicine at the University of Pennsylvania (https://www.itmat.upenn.edu/), the NIH/NHLBI National Research Service Award postdoctoral fellowship which included salary support (T32HL007843, https://www.nhlbi.nih.gov/), the Measey Foundation (https://www.measeyfoundation.org/), and the Doris Duke Foundation (https://www.dorisduke.org/). Z.A. is supported by the NIH/NHLBI (R01-HL152446, https://www.nhlbi.nih.gov/) and the Department of Defense (W81XWH18-1-0503, https://cdmrp.health.mil/funding/), which contribute salary support. The funders had no role in study design, data collection and analysis, decision to publish, or preparation of the manuscript.

**Competing interests:** I have read the journal's policy and the authors of this manuscript have the following competing interests: MGL receives research support to his institution from MyOme outside of this work. SMD receives research support to his institution from RenalytixAI and in kind support from Novo Nordisk, both outside of this work.

$P<0.001$). These results were consistent when considering the effect of change in fraction of similarity to the African reference population by ADMIXTURE as a continuous variable. Similar results were observed for the effect of TTNtvs on Afib and LVEF. Our findings demonstrate that TTNtvs are associated with increased risk of DCM, reduced LVEF, and Afib among a diverse cohort. There is no significant difference in effect of TTNtvs across fractions of similarity to the AFR reference population suggesting genetic background should not be considered when screening individuals for titin-related cardiovascular disease.

## Author summary

Variants in several different genes have been identified as causal changes that predispose individuals to cardiovascular disease. Within the context of dilated cardiomyopathy (DCM), variants in the *TTN* gene (the gene that encodes the Titin protein which is the largest protein in the human body) are the most well-known genetic changes that contribute to an individual's risk of DCM. However, investigations into the effect of *TTN* variants have suggested that there is a smaller effect among individuals genetically similar to the African reference populations. This has led some to consider how ancestry may impact the risk conferred by variants in *TTN*. However, more recently evidence has suggested that the discrepancy observed between different ancestries is related more to sample size and that most studies of genetic risk factors for DCM have been carried out in largely homogenous populations consisting primarily of individuals genetically similar to European reference populations. Here we observe that the effects of variants in *TTN* are similar across populations genetically similar to European and African reference populations either when individuals are dichotomized by population subgroup similarity or when genetic similarity is considered as a continuous variable.

## Introduction

Titin, the protein encoded by the *TTN* gene, is the largest protein in the human body. It is found in the sarcomere where it spans from the Z-disk to the M-band, and is critical for sarcomere assembly, contraction and relaxation in striated cardiac muscle [1]. In populations largely composed of individuals genetically similar to the 1000 Genomes Project (1000G) [2] European reference population (EUR), heterozygous *TTN* truncating variants (TTNtvs) that encode for shortened forms of the titin protein have been identified as a common genetic cause of dilated cardiomyopathy (DCM). These TTNtv have been associated with 25% of familial cases of DCM, and 10–20% of sporadic cases [3–8]. Only variants located in exons that are highly likely to be spliced into adult cardiac *TTN* transcripts, known as high percentage spliced in (hiPSI) variants, are pathogenic [4]. hiPSI TTNtvs may cause DCM by reducing abundance of full-length TTN protein (haploinsufficiency) and/or through dominant negative effects [9,10].

We previously demonstrated the association of hiPSI TTNtvs with DCM exclusively in EUR individuals, but were unable to detect an association between hiPSI TTNtvs and DCM among Penn Medicine Biobank (PMBB) participants genetically similar to the 1000G African reference population (AFR; odds ratio [OR] 1.8, 95% CI 0.2 to 13.7, $P = 0.57$), or among AFR participants of the Jackson Heart Study (S1 Table) [8]. Recently, a case-control analysis of individuals with DCM demonstrated a statistically significant but attenuated effect of predicted loss-of-function (pLOF) variants in *TTN* among AFR individuals compared to EUR individuals, potentially underscoring the limitations of applying genetic understanding of disease derived from a single ancestry group to diverse populations [11]. However, a separate study of hiPSI TTNtv carriers undertaken among *All of Us* participants showed a similar effect estimate among individuals genetically similar to either the EUR or AFR reference population (S1 Table) [12].

Common and rare genetic risk factors for disease have historically been analyzed among different populations separated by ancestry [13–16]. This reflects the belief that the artificial grouping of individuals into genetically common cohorts allows improved assignment of genetic risk. However, greater understanding of genetic similarities between groups of individuals combined with the results of genetic admixture within different populations suggests that instead of strict dichotomization into groups, genetic ancestry is better thought of as a continuum of relative similarity [14].

Here we describe the assessment of DCM risk and reduced left ventricular ejection fraction (LVEF) conferred by hiPSI TTNtvs in a large, diverse biobank. We investigate this relationship either when individuals are stratified by "population group" according to their genetic similarity to one of the 1000G continental-level reference populations or using ADMIXTURE to treat genetic similarity as a continuous variable. Due to previous research identifying TTNtvs risk factors for atrial fibrillation (Afib) [17–19], we performed a similar investigation into the link between TTNtvs and Afib.

## Results

### Study population

There were 43,731 participants in the analytic cohort (**Table 1**). The median age at analysis was 57 years (IQR: 45–69 years) and 21,907 (50%) were female. One percent (436 individuals) carried a hiPSI TTNtv, 1,112 individuals (2.5%) had a DCM diagnosis, and 4,920 individuals (11%) had an Afib diagnosis. There were 365 unique TTNtvs identified including 174 stop-gains, 44 frameshifting insertion-deletions, and 147 essential splice site variants.

**Table 1. Clinical characteristics of individuals in the Penn Medicine Biobank with and without high percentage spliced-in *TTN* truncating variants.**

| Demographic | | hiPSI TTNtv (436) | non-hiPSI TTNtv (43,295) | P-value Difference |
|---|---|---|---|---|
| Age, median (IQR) | | 56 (45-67) | 57 (45-69) | 0.65 |
| Female Sex, N (%) | | 184 (42%) | 21,723 (50%) | <0.001 |
| "Genetically Similar" Population Group, N (%) | | | | |
| | EUR | 330 (75%) | 29,626 (68%) | 0.002 |
| | AFR | 88 (20%) | 11,048 (26%) | 0.008 |
| | AMR | 2 (0.4%) | 564 (1.3%) | 0.12 |
| | EAS | 4 (0.9%) | 466 (1.1%) | 0.74 |
| | SAS | 5 (1.1%) | 554 (1.3%) | 0.79 |
| | Unknown | 7 (1.6%) | 1,037 (2.4%) | 0.28 |
| Dilated Cardiomyopathy, N (%) | | 52 (12%) | 1,060 (2.4%) | <0.001 |
| Ischemic Cardiomyopathy, N (%) | | 64 (15%) | 5,305 (12%) | 0.13 |
| Atrial Fibrillation, N (%) | | 89 (20%) | 4,831 (11%) | <0.001 |

PLOS Genetics

## Genetic diversity analysis

The vast majority of understanding of the genetic risk of DCM, including that conferred by TTNtvs, was derived from investigation of cohorts primarily comprised of individuals genetically similar to the EUR reference population. To determine the extent of genetic diversity in this PMBB cohort, we performed principal component analysis of PMBB participants (Fig 1A-1C) using principal components (PC) derived from the continental-level reference populations (Fig 1D-1F). To further characterize population as a continuous variable, we performed ADMIXTURE analysis on all PMBB participants S1 Fig) which demonstrated the range of admixed population in PMBB. Notably, variances in percentage similarity to specific populations may overestimate overall fractions of similarity in this plot due to ADMIXTURE's optimization algorithm terminating at a local maximum of the likelihood as described previously [20].

## Effect of TTNtvs on risk of DCM across EUR and AFR populations groups

We previously demonstrated a significant effect on DCM risk conferred by carrying TTNtvs among individuals genetically similar to the EUR reference population, however we could not detect an effect among individuals genetically similar to the AFR reference population [8]. In this updated study, encompassing a substantially larger number of participants, we evaluated the effect of TTNtvs among individuals separated into population groups to compare to our previous published results. Carrying a hiPSI TTNtv was associated with DCM among individuals genetically similar to the EUR and AFR reference populations (**Fig 2A**, EUR: OR=6.12, 95% CI 4.33 to 8.65, $P<0.001$, AFR: OR=3.44, 95% CI 1.97 to 5.99, $P<0.001$, Meta: OR=5.21, 95% CI 3.88 to 6.99, $P<0.001$; heterogeneity $\chi^2=2.97$, $P=0.08$, $I^2=66\%$). A Z-test showed no statistical difference between effects ($P=0.40$). These results were similar when individuals with ischemic cardiomyopathy were excluded from the analysis, however there was increased evidence of heterogeneity in part because of the decreased sample size (S2 Fig).

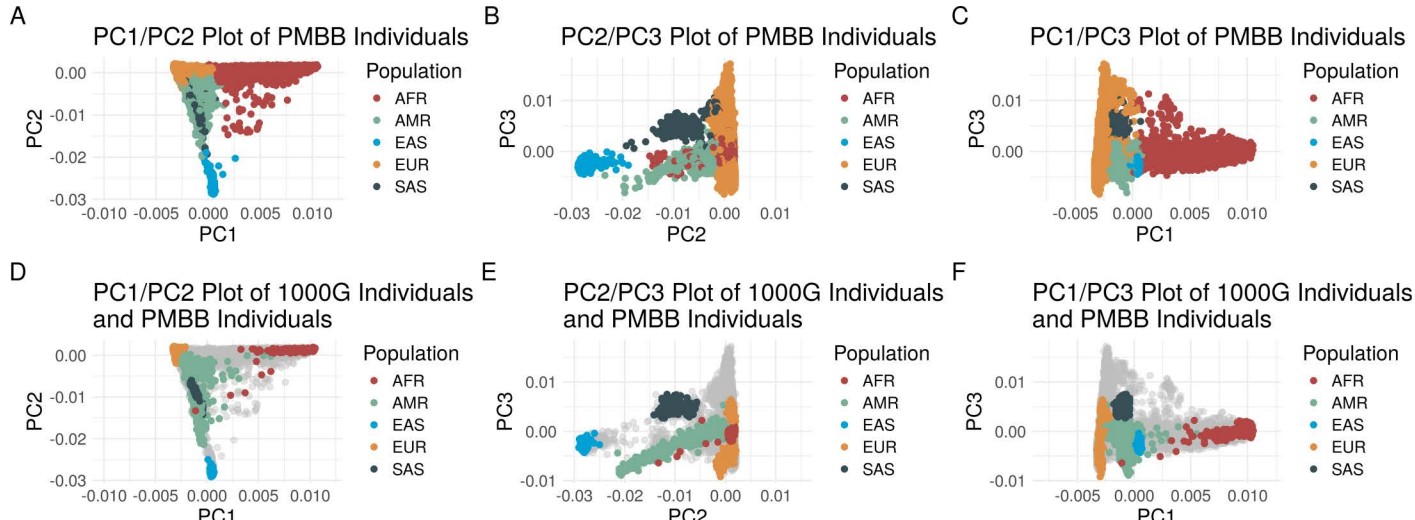

**Fig 1. Principal component based clusters of individuals in the Penn Medicine Biobank compared to the continental-level clusters of 1000 Genomes Project individuals demonstrating the degree of overlap of different genetically similar groups. A-C:** Discrete labelling of the position of each individual within the Penn Medicine Biobank (PMBB) colored by genetically similar group based on 1000 Genomes Project (1000G) continental-level reference populations in a plot of (**A**) principal component (PC) 1 versus PC2, (B) PC2 versus PC3, and (C) PC1 versus PC3. D-F: Discrete labelling of the position of each 1000G reference panel individuals colored by continental region with PMBB individuals represented by grey dots in a plot of (D) PC1 versus PC2, (E) PC2 versus PC3, and (F) PC1 versus PC3.

A

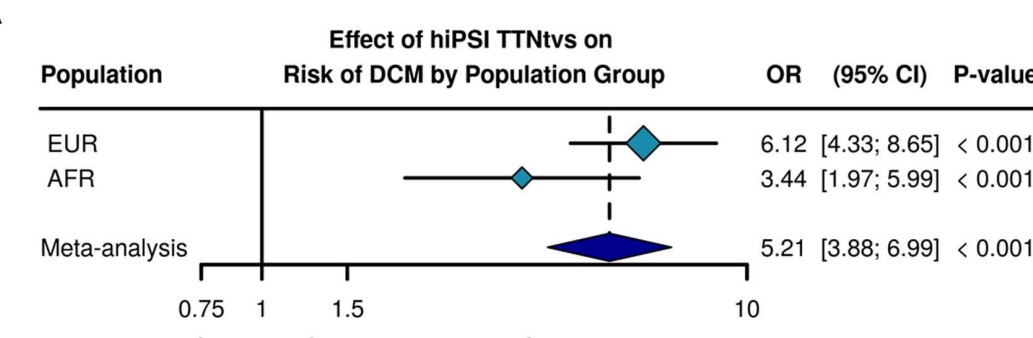

B

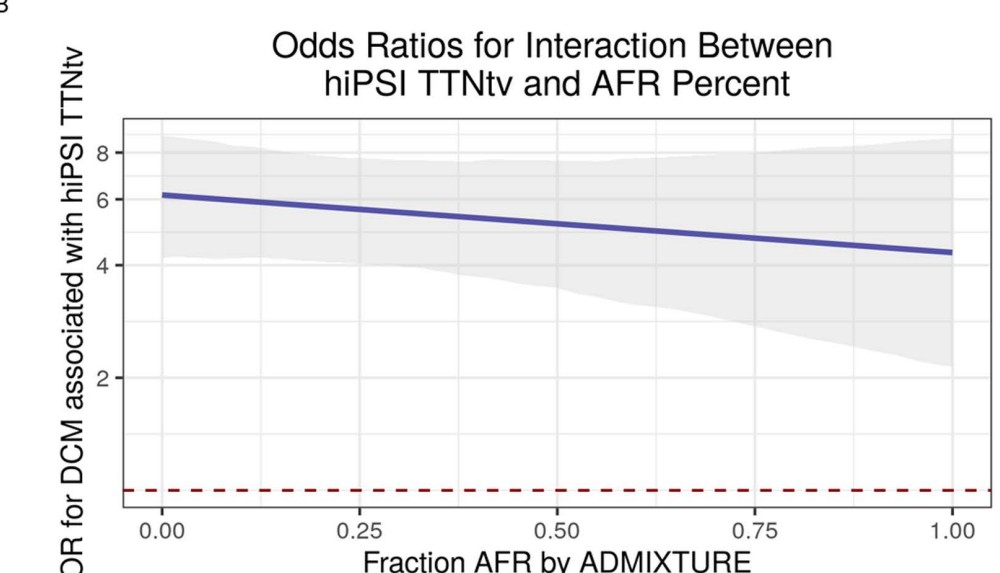

**Fig 2. Effect of *TTN* truncating variants on risk of dilated cardiomyopathy, stratified by genetic similarity to 1000 Genomes Project reference population and by fraction AFR by ADMIXTURE.** (A) Logistic regression analysis of the association between hiPSI TTNtvs on risk of DCM stratified by genetic similarity to the EUR and AFR reference populations with heterogeneity statistics included. (B) Logistic regression analysis of the interaction between hiPSI TTNtv and fraction AFR by ADMIXTURE where the dark blue line is the interaction effect estimate across the continuum of fraction of AFR, the grey shaded area are the 95% CIs, and the red dashed line is OR = 1. OR = odds ratio; CI = confidence interval; EUR = European reference population; AFR = African reference population.

To further characterize the effect of TTNtvs across different populations groups, we modeled DCM as described by the interaction between hiPSI TTNtvs and proportion AFR by ADMIXTURE along with covariates including age, sex, and the first 5 genetic principal components (S2 Table). As shown in **Fig 2B**, the effect of hiPSI TTNtv on DCM risk is consistent across the spectrum of AFR fraction, and the hiPSI effect estimate was not significantly associated with the fraction of genetic similarity to AFR ($P = 0.43$). Notably, the 95% confidence intervals increase along with AFR fraction suggesting that analysis of this relationship remains limited by sample size among those with the highest proportion of AFR genetic similarity even with our increased cohort population. Taken together, these results suggest that TTNtvs are associated with DCM across EUR and AFR population groups, and specifically, there is no significant difference in hiPSI TTNtv effect on DCM risk related to genetic similarity to the AFR reference population either when considered as a dichotomized factor or continuous proportion.

## Effect of TTNtvs on minimum LVEF across EUR and AFR populations groups

We next sought to determine the effect of hiPSI TTNtvs on minimum LVEF. Using transthoracic echocardiography (TTE) data from all PMBB participants with LVEF values recorded, we found that carrying a hiPSI TTNtv was associated with a lower minimum LVEF among individuals genetically similar to the EUR and AFR reference populations (**Fig 3A**, EUR: Beta=-10.16%, 95% CI -12.80% to -7.52%, *P*<0.001, AFR: Beta=-12.85%, 95% CI -18.32% to -7.37%, *P*<0.001, Meta: Beta=-10.67%, 95% CI -13.04% to -8.29%, *P*<0.001; heterogeneity χ²=0.75, *P*=0.39, I²=0%). A Z-test showed no statistical difference between effects across population groups (*P*=0.19). These results were similar when individuals with ischemic cardiomyopathy were excluded from the analysis (S3 Fig)

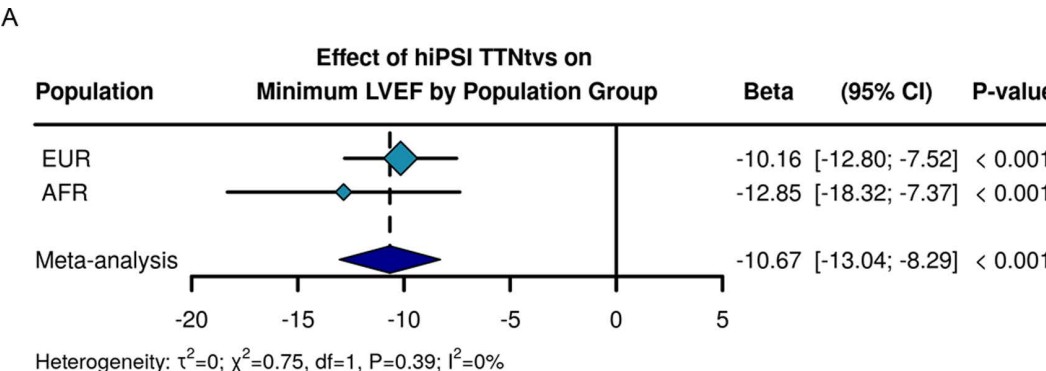

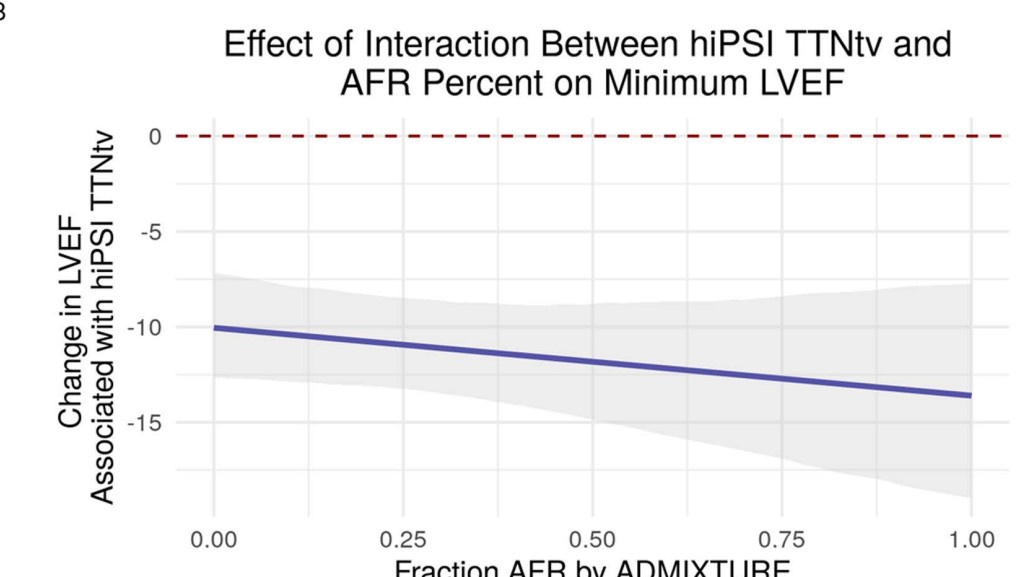

**Fig 3. Effect of high percentage spliced in *TTN* truncating variant on minimum left ventricular ejection fraction reduction stratified by genetic similarity to 1000 Genomes Project reference population and by fraction AFR by ADMIXTURE.** (A) Linear regression analysis of the association between hiPSI TTNtvs on risk of decreased minimum LVEF stratified by genetic similarity to the EUR and AFR reference populations with heterogeneity statistics included. (B) Linear regression analysis of the interaction between hiPSI TTNtv and fraction AFR by ADMIXTURE where the dark blue line is the interaction effect estimate across the continuum of fraction of AFR, the grey shaded area are the 95% CIs, and the red dashed line is Beta=0. CI=confidence interval; EUR=European reference population; AFR=African reference population; LVEF=left ventricular ejection fraction.

We then modeled minimum LVEF as described by the interaction between hiPSI TTNtvs and proportion AFR by ADMIXTURE along with covariates including age, sex, and the first 5 genetic PCs (S3 Table). The effect estimate is not significantly associated with an interaction between hiPSI and AFR fraction (*P* = 0.32) (**Fig 3B**). Altogether, these results suggest that TTNtvs are associated with lower minimum LVEF across EUR and AFR population groups, and that this effect estimate is not impacted by a change in fraction of genetic similarity to the AFR reference population.

### Effect of TTNtvs on risk of atrial fibrillation across EUR and AFR populations groups

To test the effect of TTNtvs on risk of Afib, another cardiovascular diagnosis previously shown to associate with TTNtvs, we evaluated the effect of TTNtvs among individuals separated into population groups similar to above. Carrying a hiPSI TTNtv was associated with an increased risk of Afib among individuals genetically similar to the EUR and AFR reference populations with (S4A Fig, EUR: OR=2.08, 95% CI 1.59 to 2.72, *P* < 0.001, AFR: OR=2.01, 95% CI 1.04 to 3.88, *P* = 0.04, Meta: OR=2.07, 95% CI 1.61 to 2.65, *P* < 0.001; heterogeneity $\chi^2$ = 0.01, *P* = 0.93, $I^2$ = 0%). A Z-test showed no statistical difference between effects across population groups (*P* = 0.96). These results were similar when individuals with ischemic cardiomyopathy were excluded from the analysis, though the effect estimates were somewhat attenuated (S4B Fig). Finally, we modeled Afib as described by the interaction between hiPSI TTNtvs and proportion AFR by ADMIXTURE along with covariates including age, sex, and the first 5 genetic PCs (S4 Table). S4C Fig demonstrates that the hiPSI effect estimate was not significantly associated with the fraction of genetic similarity to AFR (*P* = 0.22). Taken together, these results suggest that TTNtvs are associated with Afib across EUR and AFR population groups.

## Discussion

In this project, we present our updated analysis of the effect of TTNtvs in the PMBB. Our observations demonstrated that hiPSI TTNtvs associated with increased risk of DCM, reduced minimum LVEF, and Afib across EUR and AFR population groups when individuals were dichotomized by genetic similarity to reference populations. These findings were supported by our results using ADMIXTURE analysis to show robust effects of TTNtvs independent of fraction of AFR population group similarity. Our findings support the conclusion that hiPSI TTNtvs affect risk of heritable cardiac disease to a similar magnitude across EUR and AFR population groups.

The question of whether and to what extent TTNtvs affect cardiovascular disease risk among individuals genetically similar to the AFR reference population provided an opportunity to reassess the role genetic background plays in how we approach individual risk in cardiovascular disease. Although genetic epidemiology studies have historically relied on arbitrary cutoffs to differentiate and label populations of similar ancestry, genetic ancestry is more appropriately considered as a continuum due to the degree of admixture even among similar populations [21–23]. When proportion of AFR population similarity as determined by ADMIXTURE was utilized as a continuous variability in the present analysis, there was a consistent effect of hiPSI TTNtvs across the analytic cohort. This result reduces the emphasis placed on dichotomous genetic similarity especially in a substantially admixed population. Future evaluation of monogenic risk of cardiovascular disease may benefit from utilizing a similar approach to integrate population group analysis as a continuous variable, which may be more broadly applicable than analyses using genetically similar group dichotomization alone.

These results refine our previous findings that hiPSI TTNtvs are associated with DCM among individuals genetically similar to EUR reference populations in PMBB. Previously, we reported an OR of 18.7 (95% CI 9.1 to 39.4) for DCM among EUR individuals with hiPSI TTNtvs [8]. The range of effect estimates in our analysis of risk by fraction of AFR similarity, and the effect estimate for individuals genetically similar to EUR reference panels (OR=6.12, 95% CI 4.33 to 8.65, *P* < 0.001) was reduced compared to this prior estimate. The attenuation of effect we observed between our initial publication and the present study may reflect a winner's curse in our initial study, where the initially-reported effect size tends to be over-estimated in genetic association studies [24]. However, the makeup of PMBB has also changed substantially in the past six years such that it is less enriched for cardiovascular disease (6.2% prevalence of DCM previously compared

to 2.5% now) due to increasingly broad regional enrollment from centers of primary care which may have had a more meaningful impact on our current results.

Our findings also clarify the effect of hiPSI TTNtvs in individuals genetically similar to the AFR reference population. We previously identified no statistically significant effect among PMBB participants genetically similar to the AFR reference population [8]. This corroborated other evidence suggesting the possibility of differing effects across diverse populations [4]. Notably, our previous PMBB cohort included only 2,123 participants genetically similar to the AFR reference population, 20 of whom carried a hiPSI TTNtv, suggesting we may have been previously underpowered to identify an effect similar to what we presently report. We now identify a significant effect of hiPSI TTNtvs (OR=3.44, 95% CI 1.97 to 5.99, $P<0.001$) in this population.

These findings are similar to a recently published manuscript demonstrating an equivalent effect of TTNtvs in individuals genetically similar to AFR and EUR reference populations in *All of Us* [12]. In their report, Shetty, et al, show that DCM, heart failure, and Afib risk associated with hiPSI TTNtvs were comparable between the population groups. Our work validates these findings in a secondary cohort providing additional support for the concept that the causal effect of pathogenic variants is consistent across populations. In contradistinction to the Shetty publication, however, our analyses also consider genetic diversity on a continuous scale. It is well recognized that human genetic diversity exists on a continuous spectrum and there is frequently as much diversity within genetically similar populations as there is between them [25]. The dichotomization of genetic diversity into subpopulations can be problematic as it fails to acknowledge this concept. Furthermore, the tendency to conflate genetic subpopulations with the social construct of race and ethnicity is problematic. Including measures of genetic diversity as a continuous covariate creates a more flexible analytic framework and avoids these pitfalls. In the current analysis it permits the firm conclusion that TTNtv effects DCM risk across EUR and AFR population groups.

## Limitations

This work has some limitations. First, reliance on electronic health records to establish a diagnosis of DCM may have misclassified a subset of individuals. Second and similarly, TTEs were not interpreted by a uniform reader. Third, our cohort is also largely composed of EUR and AFR individuals, limiting generalizability to other genetically similar groups. Finally, we did not perform a centroid bias adjustment in the PMBB PC projection that may have resulted in under-adjustments for differences in population.

## Conclusion

In summary, our updated analysis demonstrates a strong association between hiPSI TTNtvs and DCM across a range of genetic backgrounds, including subjects genetically similar to African reference population. These data suggest that recommendations for genetic testing and counseling should not differ between individuals genetically similar to EUR or AFR reference populations.

## Methods

### Ethics statement

This study utilizing PMBB data is approved by the University of Pennsylvania Institutional Review Board (IRB) under protocol# 813913. Written consent was provided by all study participants.

### Study population

The Penn Medicine BioBank is a genomic and precision medicine cohort comprising participants who receive care in the Penn Medicine health system and who consent to linkage of electronic health records with biospecimens, including 43,731 with DNA which has undergone whole exome sequencing (pmbb.med.upenn.edu). As previously described [8],

DCM was defined either as ≥2 outpatient or ≥1 inpatient encounters with: 1) the *International Classification of Diseases, 10th Revision* (ICD10) diagnosis code of I42.0; or 2) ICD10 codes I42.8 or I42.9 or the *International Classification of Diseases, Ninth Revision* (ICD9) codes 425.4, 425.8, or 425.9, and mention of "dilated cardiomyopathy" or "DCM" in free text encounter notes. Ischemic cardiomyopathy (ICM) was defined as ≥2 outpatient or ≥1 inpatient encounters with ICD10 codes I24 or I25, or ICD9 codes 411 or 414. Afib was defined as ≥2 outpatient or ≥1 inpatient encounters with ICD10 codes I48, I48.1, I48.2, or I48.9, or ICD9 codes 427.2 or 427.21. A total of 9,020 individuals in PMBB had transthoracic echocardiography (TTE) data available that included left ventricular ejection fraction (LVEF).

## Genetic data

Whole exome sequencing was performed as previously described by Regeneron Genomics Center [26], Individual patient DNA samples were processed and sequenced on the Illumina NovaSeq 6000 (Albany, NY, USA). WeCall variant caller (v2.0.0, https://github.com/Genomicsplc/wecall) was employed for sequence alignment (GRCh38), variant identification, and genotype assignment as previously described [27]. Quality control exclusions included sex errors, high rates of heterozygosity (D-statistic > 0.4), low sequence coverage (<85% of targeted bases achieving 20X coverage), and genetically identified sample duplicates. Single nucleotide variants (SNVs) were filtered for a read depth 7 and were retained if they either had at least one heterozygous variant genotype with an allele balance ratio 0.15, or a homozygous variant genotype [28]. Insertion-deletion variants (INDELs) were filtered for a read depth 10 and either a heterozygous variant genotype with an allele balance 0.20, or a homozygous variant genotype. Using PLINK (version 2.0) [29], we identified and excluded cryptically related individuals by randomly selecting one relative from each pair/group and excluding all others (pi-hat > 0.25) through identity by descent (IBD) analysis using the equation pi-hat = $P$(IBD = 2) + 0.5 × $P$(IBD = 1) where $P$(IBD = 2) is the probability of sharing two alleles identical by descent and $P$(IBD = 1) is the probability of sharing one allele identical by descent.

## Population group descriptors

To compare the effect of TTNtvs across African (AFR) and European (EUR) populations, we conducted principal component analysis (PCA) on genetic data from PMBB participants using common variants. We employed the fastmode setting in smartpca (https://github.com/DReichLab/EIG) to generate 20 eigenvectors and projected PMBB participants onto HapMap3 (HM3) reference PC space. Population assignment utilized Kernel Density Estimation (KDE) with the KernSmooth package (version 2.23-26) as previously described [27]. We constructed KDE models for each HM3 population using a 300 × 300 grid for both PC1/PC2 and PC3/PC4 dimensions. Bandwidth parameters were determined using the default "oversmoothed bandwidth selector" as described by Wand and Jones [30]. For the PC1/PC2 KDE, the bandwidths by HM3 population included GIH (Gujarati Indians in Houston, Texas) = 0.007, CEU (Utah residents with Northern and Western European ancestry from the CEPH collection) = 0.008, ASW (African ancestry in Southwest USA) = 0.017, TSI (Toscans in Italy) = 0.001, MEX (Mexican ancestry in Los Angeles, California) = 0.006, and everything else = 0.008. For the PC3/4 KDE, the bandwidths by HM3 population included GIH = 0.0225, CEU = 0.004, ASW = 0.005, TSI = 0.004, MEX = 0.013, and everything else = 0.003. PMBB participants were assigned to population groups based on their highest likelihood scores relative to the HM3 reference populations which were subsequently mapped to 1000G continental-level reference populations. While this approach is imperfect as it reinforces the artificial discretization of genetic diversity, it is necessary to create groups of individuals that share a component of common genetic variation in order to facilitate the evaluation of TTNtvs in these diverse populations.

## ADMIXTURE analysis

To quantify differences in genetic similarity to reference populations as a continuous variable among PMBB participants, proportional population group estimates were performed using ADMIXTURE [31]. ADMIXTURE calculates the proportion of genetic similarity to reference populations using genotype data. These data are then algorithmically assessed based on

the number of assumed reference populations provided to produce a proportion for each individual in the analytic cohort. In the present analysis, we performed a supervised ADMIXTURE analysis where 1000G individuals were labeled with their respective corresponding continental-level population group. Learning from those labeled 1000G individuals, fractions of continental-level population similarity were then assigned to PMBB participants using ADMIXTURE default setting and k (latent populations) = 5.

### TTN truncating variant calls

TTNtvs were filtered for minor allele frequency < 0.001 and selected based on predicted loss of function, truncating variants using ANNOVAR [32]. Splice site variants were screened for those affecting canonical donor or acceptor splice sites (two bases flanking either exon). Variants were considered hiPSI if percent spliced in was greater than 90% (PSI > 0.9) [4].

### Statistical analysis

Logistic regression was employed to evaluate the association of hiPSI TTNtvs with prevalent DCM adjusting for age-at-analysis, sex, and the first five genetic PCs. We utilized Firth's bias-reduced logistic regression to model this effect due to the relative rarity of events using the logistf package (version 1.26.0) in R [33]. Linear regression was used to assess the effect of hiPSI TTNtvs on minimum left ventricular ejection fraction (LVEF) determined prior or subsequent to enrollment, adjusting for age-at-analysis, sex, and the first five genetic PCs. Logistic and linear regression analyses of the effect of hiPSI TTNtvs were completely stratified by genetically similar population group, specifically EUR and AFR, and the population group hiPSI TTNtv effects were meta-analyzed using the inverse variance weighted method with fixed effects. Logistic and linear regression analyses of the effect of AFR proportion based on ADMIXTURE results, including the assessment of the interaction term between hiPSI TTNtv and AFR proportion, were not stratified by population group but only included EUR and AFR individuals included in the meta-analyzed analytic cohort. Plots for the interaction of hiPSI TTNtv and proportion AFR were created using the *interplot* (version 0.2.3) package. *Interplot* is a user-friendly way to visualize changes in one variable in a two-way interaction term that is conditional on the value of another included variable a single regression statement. Heterogeneity between effect estimates was evaluated using $\chi^2$, $I^2$, and $\tau^2$ tests. A formal z-test was also performed to detect the difference between effect estimates across population groups. Statistical analyses were performed using R (version 4.2.3). Code used in these analyses will be made publicly available at the time of publication.

### Supporting information

**S1 Table. Summary of results from previously published cohort studies on the association of *TTN* truncating variants and prevalent dilated cardiomyopathy diagnosis. hiPSI TTNtv = high percentage spliced in titin truncating variants; DCM = Dilated cardiomyopathy; EUR = genetically similar to the 1000 Genomes Project European reference population; AFR = genetically similar to the 1000 Genomes Project African reference population; OR = odds ratio.**
(XLSX)

**S2 Table. Effect of hiPSI *TTN* truncating variants, a 0.1 change in fraction of genetic similarity to the AFR reference population, and the interaction of the two covariates in logistic regression analysis of DCM risk accounting for age, sex, and the first five genetic PCs. hiPSI = high percentage spliced in; DCM = Dilated cardiomyopathy; UCB = upper confidence bound; LCB = lower confidence bound; AFR = genetically similar to the 1000 Genomes Project African reference population.**
(XLSX)

**S3 Table. Effect of hiPSI *TTN* truncating variants, a 0.1 change in fraction of genetic similarity to the AFR reference population, and the interaction of the two covariates in linear regression analysis of change in LVEF**

accounting for age, sex, and the first five genetic PCs. hiPSI=high percentage spliced in; LVEF=left ventricular ejection fraction; UCB=upper confidence bound; LCB=lower confidence bound; AFR=genetically similar to the 1000 Genomes Project African reference population.
(XLSX)

**S4 Table. Effect of hiPSI *TTN* truncating variants, a 0.1 change in fraction of genetic similarity to the AFR reference population, and the interaction of the two covariates in logistic regression analysis of Afib risk accounting for age, sex, and the first five genetic PCs. hiPSI=high percentage spliced in; Afib=Atrial fibrillation; UCB=upper confidence bound; LCB=lower confidence bound; AFR=genetically similar to the 1000 Genomes Project African reference population.**
(XLSX)

**S1 Fig. ADMIXTURE population group analysis in the Penn Medicine Biobank cohort.** Vertical bars represent individual participants with stacked color bars representing a particular individual's fractional composition of 1000 Genomes Project continental-level reference population groups (AFR: African reference population; AMR: Americas reference population; EAS: east Asian reference population; EUR: European reference population, SAS: south Asian reference population).
(TIFF)

**S2 Fig. Effect of high percentage spliced in titin truncating variant on risk of dilated cardiomyopathy diagnosis stratified by genetically similar group excluding those with ischemic cardiomyopathy.** Logistic regression analysis of the association between hiPSI TTNtvs and DCM diagnosis among individuals genetically similar to the 1000 Genomes Project European and African reference population, and meta-analyzed. OR = odds ratio; CI=confidence interval; EUR=individuals genetically similar to the European reference population; AFR=individuals genetically similar to the African reference population; ICM=ischemic cardiomyopathy.
(TIFF)

**S3 Fig. Effect of high percentage spliced in titin truncating variant on minimum LVEF stratified by genetic similarity to 1000 Genomes Project reference population excluding those with ischemic cardiomyopathy.** Linear regression analysis of the association between hiPSI TTNtvs on risk of decreased minimum LVEF stratified by genetic similarity to the 1000G EUR and AFR reference populations. CI=confidence interval; EUR=individuals genetically similar to the European reference population; AFR=individuals genetically similar to the African reference population; ICM=ischemic cardiomyopathy.
(TIFF)

**S4 Fig. Effect of *TTN* truncating variants on risk of atrial fibrillation, stratified by genetic similarity to 1000 Genomes Project reference population and by fraction AFR by ADMIXTURE.** Logistic regression analysis of the association between hiPSI TTNtvs on risk of Afib stratified by genetic similarity to the 1000G EUR and AFR reference populations including (A) and excluding (B) individuals with ischemic cardiomyopathy with heterogeneity statistics included; (C) Logistic regression analysis of the interaction between hiPSI TTNtv and fraction AFR by ADMIXTURE where the dark blue line is the interaction effect estimate across the continuum of fraction of AFR, the grey shaded area are the 95% CIs, and the red dashed line is OR = 1. OR = odds ratio; CI=confidence interval; EUR=1000G European reference population; AFR=1000G African reference population; ICM=ischemic cardiomyopathy.
(TIF)

**S1 Text. List of the Penn Medicine BioBank Banner team members with their contributions.**
(DOCX)

## Acknowledgments

We acknowledge the Penn Medicine BioBank (PMBB) for providing data and thank the patient-participants of Penn Medicine who consented to participate in this research program. We would also like to thank the Penn Medicine BioBank team and Regeneron Genetics Center for providing genetic variant data for analysis.

**Penn Medicine BioBank Banner team members:**

Daniel J. Rader, Marylyn D. Ritchie, JoEllen Weaver, Nawar Naseer, Afiya Poindexter, Khadijah Hu-Sain, Yi-An Ko, Meghan Livingstone, Fred Vadivieso, Stephanie DerOhannessian, Teo Tran, Julia Stephanowski, Monica Zielinski, Ned Haubein, Joseph Dunn, Anurag Verma, Colleen Morse Kripke, Marjorie Risman, Yuki Bradford, Scott Dudek, Theodore Drivas.

The full list of the group team members with their contributions is provided in S1 Text.

## Author contributions

**Conceptualization:** John DePaolo, Marc R Bornstein, Shefali S. Verma, Michael G. Levin, Zoltan Arany, Scott Damrauer.

**Data curation:** John DePaolo, Renae Judy, Shefali S. Verma.

**Formal analysis:** John DePaolo, Marc R Bornstein, Renae Judy, Michael G. Levin, Zoltan Arany, Scott Damrauer.

**Funding acquisition:** John DePaolo, Michael G. Levin, Zoltan Arany, Scott Damrauer.

**Investigation:** John DePaolo, Marc R Bornstein, Renae Judy, Sarah Abramowitz, Michael G. Levin, Zoltan Arany, Scott Damrauer.

**Methodology:** John DePaolo, Marc R Bornstein, Renae Judy, Shefali S. Verma, Michael G. Levin, Zoltan Arany, Scott Damrauer.

**Project administration:** Scott Damrauer.

**Resources:** Shefali S. Verma, Zoltan Arany, Scott Damrauer.

**Software:** Shefali S. Verma, Scott Damrauer.

**Supervision:** Zoltan Arany, Scott Damrauer.

**Validation:** John DePaolo, Sarah Abramowitz.

**Visualization:** John DePaolo, Michael G. Levin.

**Writing – original draft:** John DePaolo.

**Writing – review & editing:** John DePaolo, Marc R Bornstein, Sarah Abramowitz, Michael G. Levin, Zoltan Arany, Scott Damrauer.

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
