## [Decision Letter · Decision Letter 0]

Dear Dr Damrauer,

Thank you very much for submitting your Research Article entitled 'Titin-Truncating variants Predispose to Dilated Cardiomyopathy in Diverse Populations' to PLOS Genetics.

The manuscript was fully evaluated at the editorial level and by independent peer reviewers. The reviewers appreciated the attention to an important problem, but raised some substantial concerns about the current manuscript. Based on the reviews, we will not be able to accept this version of the manuscript, but we would be willing to review a much-revised version. We cannot, of course, promise publication at that time.

If you decide to revise the manuscript for further consideration at PLOS Genetics, please aim to resubmit within the next 60 days, unless it will take extra time to address the concerns of the reviewers, in which case we would appreciate an expected resubmission date by email to plosgenetics@plos.org.

If present, accompanying reviewer attachments are included with this email; please notify the journal office if any appear to be missing. They will also be available for download from the link below. You can use this link to log into the system when you are ready to submit a revised version, having first consulted our Submission Checklist .

PLOS has incorporated Similarity Check , powered by iThenticate, into its journal-wide submission system in order to screen submitted content for originality before publication. Each PLOS journal undertakes screening on a proportion of submitted articles. You will be contacted if needed following the screening process.

To resubmit, log into your Editorial Manager account and select the option 'Revise Submission' in the 'Submissions Needing Revision' folder.

We are sorry that we cannot be more positive about your manuscript at this stage. Please do not hesitate to contact us if you have any concerns or questions.

Yours sincerely,

Todd L. Edwards

Guest Editor

PLOS Genetics

Xiaofeng Zhu

Section Editor

PLOS Genetics

Reviewer's Responses to Questions

**Comments to the Authors:**

Reviewer #1: Summary

======

DePaolo and colleagues used data from the population-based Penn Medicine Biobank (PMBB) to evaluate whether the effect of carrying a high percent spliced in (hiPSI) TTN truncating variant (TTNtv) on the risk of prevalent dilated cardiomyopathy (DCM) and related phenotypes varied across populations. They found that the effect of hiPSI TTNtvs on DCM risk, atrial fibrillation (Afib) risk, and left ventricular ejection fraction (LVEF) was similar across bins of genetic distance (GD), the metric that they chose to measure dissimilarity from European ancestry (EUR). Importantly, they also revised their conclusion in a prior study regarding the lack of effect of TTNtvs in African ancestry (AFR) populations, finding that hiPSI TTNtvs were associated with increased DCM and Afib risk and lower LVEF in AFR populations. While the results of this study are important, there are methodological issues that should be addressed, including undesirable properties of their GD metric, some technical statistical issues, and incomplete description of some methods. A revision addressing all issues below in which the GD metric was replaced or supplemented with continuous non-European ancestry proportions would certainly be worthy of further consideration.

Specific Comments

- The authors should provide a reference detailing the design and comparative performance of the WeCall variant caller and explain why this was chosen over more standard pipelines based on the actively developed GATK software.

- All statistical techniques described assume that individuals are unrelated, but there is no mention of whether this was verified. Please describe how cryptic relatedness between PMBB participants was assessed and addressed in analysis.

- I would argue that the GD metric of Euclidean distance from the EUR centroid does not provide a relevant measure for this application. In particular, individuals in completely different regions of the ancestry PC space could have identical Euclidean distance from the EUR centroid, which means that individuals in genetically dissimilar populations (e.g., AFR and SAS) could be indistinguishable on this metric. Because individuals from dissimilar populations that may have dissimilar allele frequencies and disease risks can fall into the same grouping, this metric cannot provide an effective adjustment for population stratification without additional adjustment for other PCs. This lumping property also means that the metric does not allow estimation of interpretable ancestry-specific effects. Supplemental Table 1 and Supplemental Figure 1 show that this is more than a theoretical problem; ancestrally diverse groups were clearly lumped into the same GD category. Given these limitations, it is not entirely clear why the authors chose this metric instead of the more interpretable metric of non-European ancestry proportions (e.g., AFR, SAS, EAS, Native American) obtained from a model such as ADMIXTURE or directly from PCA. With these, the authors could determine whether and how an increasing proportion of each non-European ancestry modified the effect of carrying a hiPSI TTNtv, similar to the approach in reference 10. As this alternative approach would allow evaluation of the effect of ancestry on a continuum while minimizing residual confounding and enhancing interpretability, the authors should report results using it.

- The authors should include a description of how they defined “genetically similar” groups.

- It would be helpful to describe the logistic regression specifications used in detail. Based on the current descriptions, I have some concerns:

1) It appears that a model including age, sex, and TTNtv carrier status was fit separately in each GD bin and the results meta-analyzed. This approach is suboptimal for two reasons. First, the number of TTNtv carriers in each bin is likely small enough to warrant use of the Firth bias correction. Second, even if the Firth correction is unnecessary, the entire model could be fit with a single logistic regression in which intercept, age, sex, and TTNtv carrier status were interacted with GD bin, which would allow use of more reliable likelihood ratio tests and profile likelihood confidence intervals rather than the Wald-based procedures used for meta-analysis. Furthermore, using a single regression would allow model simplifications such as common sex and age effects that could potentially increase power.

2) The functional form for age in the model should be described and justified.

3) The estimated odds ratios presented in Supplemental Table 2 are not interpretable because they refer to a 1-unit change in GD, which is not possible as this variable has support on [0, 0.08] based on Figure 1. The odds ratios presented would be more informative if they corresponded to a 0.01-unit change.

- For the linear regression with LVEF, it is unclear why meta-analysis was used instead of the more standard approach of fitting a single model in which intercept, age, sex, and TTNtv were interacted with GD bin. As the individual-level data are available, I would recommend the latter analysis approach to facilitate model simplification and more flexible inference.

- The above comments regarding meta-analysis versus standard generalized linear model inference also apply to the genetically similar group analysis.

- Please describe the statistical tests used to produce the p-values in Table 1.

Reviewer #2: Comments are uploaded as an attachment.

**Have all data underlying the figures and results presented in the manuscript been provided?**

Reviewer #1: **No: ** The authors state that their data cannot be made publicly available as they contain sensitive information on human subjects, which is reasonable. The authors also state that they will make the data available upon reasonable request to the corresponding author.

Reviewer #2: **No: **

PLOS authors have the option to publish the peer review history of their article (what does this mean? ). If published, this will include your full peer review and any attached files.

**Do you want your identity to be public for this peer review?** For information about this choice, including consent withdrawal, please see our Privacy Policy .

Reviewer #1: No

Reviewer #2: No

---

## [Decision Letter · Decision Letter 1]

PGENETICS-D-24-00140R1

Titin-Truncating variants Predispose to Dilated Cardiomyopathy in Diverse Populations

PLOS Genetics

Dear Dr. Damrauer,

Thank you for submitting your manuscript to PLOS Genetics. After careful consideration, we feel that it has merit but does not fully meet PLOS Genetics's publication criteria as it currently stands. Therefore, we invite you to submit a revised version of the manuscript that addresses the points raised during the review process.

Please submit your revised manuscript within 60 days Feb 18 2025 11:59PM. If you will need more time than this to complete your revisions, please reply to this message or contact the journal office at plosgenetics@plos.org. Please include the following items when submitting your revised manuscript:

We look forward to receiving your revised manuscript.

Kind regards,

Todd L. Edwards

Guest Editor

PLOS Genetics

Xiaofeng Zhu

Section Editor

PLOS Genetics

Aimée Dudley

Editor-in-Chief

PLOS Genetics

Anne Goriely

Editor-in-Chief

PLOS Genetics

**Journal Requirements:**

1) We note that your Data Availability Statement is currently as follows: "Individual-level data are not publicly available due to research participant privacy concerns; requests from accredited researchers for access to individual-level data relevant to this manuscript can be made by contacting biobank@upenn.edu"

2) Please ensure that the funders and grant numbers match between the Financial Disclosure field and the Funding Information tab in your submission form. Note that the funders must be provided in the same order in both places as well.

**Reviewers' comments:**

Reviewer's Responses to Questions

**Comments to the Authors:**

Reviewer #1: Summary

======

DePaolo and colleagues have submitted a responsive and thorough revision that completely addresses prior reviewer comments regarding their genetic distance metric. That the inferences remain unchanged is highly encouraging. While I applaud their efforts thus far, there are remaining issues that should be addressed to ensure that this potentially important work meets a high but attainable standard of rigor. I would find a revision in which these are addressed to be a compelling publication.

Major Issues

========

- Pi_hat is usually used to denote an estimate of the proportion of alleles shared IBD (pi), which is twice the kinship coefficient (phi). The kinship coefficient for first cousins is 0.0625, so pi for first cousins is 0.125, not 0.25. A pi_hat threshold of 0.25 would therefore not exclude first cousins or other third-degree relatives, which may not be stringent enough to eliminate problems due to cryptic relatedness. Moreover, pi_hat (= 2 * phi_hat) is an estimated IBD probability that will vary about its expectation of 0.125 for first cousins, potentially ranging as low as ~0.088 (see 10.1093/bioinformatics/btq559). The threshold for unrelated in this publication is phi_hat<0.044 (pi_hat<0.088); others (10.1002/gepi.21896) have used phi_hat<0.025 (pi_hat<0.05) for identifying unrelated individuals. Details on how this threshold was applied to select an unrelated subset should also be provided (e.g., PLINK algorithm). Finally, methods for obtaining an estimate of pi_hat and phi_hat should be described. Given the diverse nature of this sample, methods that account for distant relatedness due to ancestry (e.g., KING, PC-Relate) would be preferable, although methods assuming a sample from a single population would tend to overestimate relatedness (10.1016/j.ajhg.2015.11.022) and exclude unrelated individuals rather than underestimate it and include related ones.

- The description of the methods for population group descriptors is not sufficiently detailed. Kernel density estimation (KDE) is a very broad set of techniques in statistics involving many tuning parameters (e.g., bandwidth) that should be described. Presumably, KDE was used to estimate a multidimensional density for each 1000G reference population over the PC space, and the number of PCs considered (i.e., dimension of this density) should be clearly stated. How the resulting densities were used to assign a new individual to a population group should also be clearly stated. Was this simply the population for which that individual had highest density? Or was a Bayes classifier with non-uniform prior weights used? Without these details, it is difficult to evaluate the reasonableness of the overall approach for assigning individuals to ancestry groups, although the broad description seems to be on the right track and the assignments in Figure 1A-C look reasonable relative to the reference.

- Several details regarding the ADMIXTURE analysis are missing. Were models with different numbers of latent populations (K) fit and compared (10.1007/978-1-0716-0199-0_4)? Were multiple runs from different starting points for each K performed and compared as recommended (10.1007/978-1-0716-0199-0_4 and 10.1093/bioinformatics/btw327)? How was the optimal K chosen?

- While use of age at enrollment is reasonable given the constraints of the data, it does require clarification of a few items in the outcome definitions. First, did diagnostic criteria for DCM and AFib consider only encounters prior to enrollment? This would be ideal as the logistic regression model then represents an implicit proportional odds survival model for DCM/Afib risk based on current status (at enrollment) data (see 10.1201/b16248, chapter 22). Second, over what period was the minimum LVEF determined, and how was this period related to age at enrollment?

- The authors should clarify in the manuscript that regressions used for the ancestry group analysis are completely stratified with the ancestry group-specific hiPSI TTNtv effects analyzed and combined using meta-analysis (although see my comments below about the issues with continued use of this approach). At the same time, the authors should clarify in the manuscript that the regressions including the interaction term with the global AFR proportion are not stratified. This entails additional assumptions regarding common effects of age at enrollment, sex, and ancestry PCs in AFR and non-AFR ancestries, so these are not different parameterizations of the ancestry-specific effect in the same model fit to the same sample. To obtain that, the authors would need to replace the AFR global ancestry proportion with a binary AFR = 1/EUR = 0 indicator and limit the analysis to individuals in just the AFR and EUR groups. All meta-analytic tests of heterogeneity and the combined TTNtv effect could then be obtained from nested models with and without the interaction.

- If groups other than AFR and EUR are included in the model for the global ancestry proportion interaction, the TTNtv effect should also be allowed to vary by the proportions of other non-EUR global ancestries. Not doing so introduces a problem similar to the genetic distance metric in the last version of the manuscript, except now individuals who are nearly completely EUR, SAS, AMR, or EAS are all lumped into the reference group with the same assumed TTNtv effect. Figure 1A-C suggests a fair amount of admixture along other axes of ancestry, so there could be enough variation to include other non-EUR global ancestry proportions that vary in the sample (e.g., SAS, EAS, AMR) and their interactions with the TTNtv effect, which would solve the problem. Otherwise, limiting this analysis to the same combined sample as the meta-analysis of EUR and AFR groups would be a reasonable approach.

- The justification for not using Firth logistic regression in response to Reviewer 2’s comment is not entirely convincing. Prior work has shown that performance issues may arise with minor allele count <400 (10.1002/gepi.21742), a condition that applies here in the stratified meta-analysis. Using Firth correction also would not affect comparability with prior work if quasi-complete or complete separation were not an issue in this or any of these other studies. I agree with Reviewer 2 that using Firth logistic regression (and penalized likelihood inference on a unified model, as per my prior comments) would greatly increase confidence in the findings of this study with a relatively rare outcome and fairly low minor allele count.

Minor Issues

========

- The analysis in reference 10 relating to the effect of TTNtvs on DCM risk appears to have been case-control, not cross-sectional. This should be corrected.

- It would be helpful to also include a reference the cited UKB paper to show that WeCall has been widely employed.

- It would be helpful to clearly describe in the manuscript whether the PCA and ADMIXTURE analyses were done on pooled 1000G and PMBB samples or whether PMBB samples were projected onto PC and ancestry spaces learned solely from 1000G samples. The response to reviewers seems to suggest that projection was used for PCA, in which case the method for centroid bias adjustment (10.1093/bioinformatics/btaa152), if any, should be described.

- The ADMIXTURE plot in Supplemental Figure 1 raises some concerns. First, the global ancestry proportions do not appear to stack to 1 for a number of individuals. Second, it would be useful to see a plot for the 1000G reference individuals to inform the labeling used for latent subpopulations. This is especially relevant because the ACB, ASW, and AMR reference samples are representative of variable recent European-African or European-Native American(-African) admixture events in the Americas.

- The conclusions in this manuscript do not apply to diverse populations such as AMR, SAS, or EAS, as appropriately noted by the authors in the limitations. However, the authors should limit the inferences to EUR and AFR (rather than “diverse populations”) more generally throughout the manuscript.

Reviewer #2: The comments are provided as an attachment.

**Have all data underlying the figures and results presented in the manuscript been provided?**

Reviewer #1: **No: ** The authors state that their data cannot be made publicly available as they contain sensitive information on human subjects, which is reasonable. The authors also state that requests from accredited researchers for access to individual-level data relevant to this manuscript can be made by contacting biobank@upenn.edu and have committed to making the code publicly available upon publication.

Reviewer #2: **No: ** Some data are not likely to be shared, but some details on the analysis are not provided (although the authors promised to be publicly available)

PLOS authors have the option to publish the peer review history of their article (what does this mean? ). If published, this will include your full peer review and any attached files.

**Do you want your identity to be public for this peer review?** For information about this choice, including consent withdrawal, please see our Privacy Policy .

Reviewer #1: No

Reviewer #2: No

**Figure resubmission:**
---

## [Decision Letter · Decision Letter 2]

PGENETICS-D-24-00140R2

Titin-Truncating variants Predispose to Dilated Cardiomyopathy in Populations Genetically Similar to African and European Reference Populations

PLOS Genetics

Dear Dr. Damrauer,

Thank you for submitting your manuscript to PLOS Genetics. Both reviewers suggested that the manuscript is acceptable after minor a revision. Therefore, we invite you to submit a revised version of the manuscript that addresses the points raised during the review process. We expect further round of review may not be necessary but we also believe a revision is needed, given the concerns raised by the reviewers.

Please submit your revised manuscript within 30 days Apr 27 2025 11:59PM. If you will need more time than this to complete your revisions, please reply to this message or contact the journal office at plosgenetics@plos.org. Please include the following items when submitting your revised manuscript:

We look forward to receiving your revised manuscript.

Kind regards,

Todd L. Edwards

Guest Editor

PLOS Genetics

Xiaofeng Zhu

Section Editor

PLOS Genetics

Aimée Dudley

Editor-in-Chief

PLOS Genetics

Anne Goriely

Editor-in-Chief

PLOS Genetics

**Additional Editor Comments:**

Please see the attached comments from Reviewer 2 and make those changes to the paper prior to publication.

**Journal Requirements:**

In the online submission form, you indicated that "Individual-level data are not publicly available due to research participant privacy concerns; requests from accredited researchers for access to individual-level data relevant to this manuscript can be made by contacting biobank@upenn.edu". All PLOS journals now require all data underlying the findings described in their manuscript to be freely available to other researchers, either

1. In a public repository

2. Within the manuscript itself

3. Uploaded as supplementary information.

**Reviewers' comments:**

Reviewer's Responses to Questions

**Comments to the Authors:**

Reviewer #1: DePaolo and colleagues have submitted a responsive and thorough revision that addresses nearly all of my comments on the prior revision. All issues itemized below should be addressable with minor changes to the manuscript.

- Reviewer 1, Comment 1: The authors should provide the PLINK version used (perhaps it was 1.07 based on the reference) and specify the method for selecting an unrelated subset (e.g., randomly select one relative from each pair).

- Reviewer 1, Comment 2: While the additional details are helpful, the authors should also provide information on the KDE software implementation, kernel type, and bandwidth selection technique.

- Reviewer 1, Comment 5: The addition in lines 379-85 mostly addresses the comment, but I believe that this clarification should apply to both logistic *and linear* regressions. Please make that correction. Also, please specify the meta-analytic method used (e.g., inverse variance weighted fixed effects).

- Reviewer 1, Comment 8: This still needs to be corrected. While the study was cross-sectional with respect to the probands, all of whom had DCM, the supplemental results on the association of TTNtvs with DCM risk in eAppendix 2 were based on a comparison of probands (cases) to *external population controls* from 1000 Genomes. As all probands in the cross-sectional sample analyzed in the main text had DCM, there would be no way to evaluate the association of TTNtvs with DCM risk in just probands; doing so would require pairing external controls with these cases, as in eAppendix 2. The authors of reference 11 also make explicit references to the case-control nature of the TTNtv analysis and its reliance on external controls in eAppendix 2 with statements such as:

-- “The odds of harboring a TTN pLOF variant were modeled as a function of case status (DCM proband or 1000 Genomes control)…”

-- “This retrospective formulation of the logistic regression was chosen because it more closely reflects the sampling design, which was stratified on both case status and ancestry so that only whether an individual was found to harbor a TTN pLOF variant was an unrestricted random quantity.”

- Reviewer 1, Comment 10: The lack of centroid bias adjustment in PC projection may result in under-adjustment for ancestry differences (10.1093/bioinformatics/btaa152, 10.1093/bioinformatics/btaa520), although shrinkage may be relatively small for the top 4-5 PCs when using the entire 1000 Genomes reference (10.1093/bioinformatics/btaa152, 10.1093/bioinformatics/btaa520). I would suggest that this be included as a potential limitation mitigated by the likelihood that shrinkage was small based on data in the references above.

- Reviewer 1, Comment 11: The corrected ADMIXTURE plot seems to indicate a degree of genetic similarity with SAS reference populations much higher than that reported by the authors in Table 1 and elsewhere for the entire cohort (10.3390/jpm12121974), all concentrated within individuals with low genetic similarity to AFR. The source of this phenomenon should be investigated and explained to the reader. One possible explanation is that ADMIXTURE's optimization algorithm terminated at a local maximum of the likelihood; the authors should try rerunning ADMIXTURE at least 10 times using different seeds and comparing the results to confirm that this is indeed a global maximum, as recommended in my prior comments and by the ADMIXTURE authors themselves (10.1007/978-1-0716-0199-0_4).

- Supplemental Tables 2 - 4: The AFR variable is the fraction of genetic similarity to the AFR reference population, which has support on [0, 1]. Because the coefficient on AFR and the interaction with hiPSI are effects for a 1-unit change in AFR, they are therefore a bit hard to interpret. The authors could rescale these coefficients to be for a change of 0.1 in the fraction of genetic similarity, which would be more interpretable without changing any inferences or plots.

- Lines 168-70, 192-3, 213-4: The phrase “…the effect estimate is not significantly associated with an interaction between hiPSI and AFR…” is a bit confusing. The authors should say that either (1) there was no evidence of interaction between hiPSI and AFR or (2) the hiPSI effect estimate was not significantly associated with the fraction of genetic similarity to AFR.

Reviewer #2: Based on this revision, I do not have major issues and would like to suggest the manuscript is publishable. Minor comments are provided as follows for improving readability.

Minor comments

1) Line 159, “P=008” seems to be typo.

2) Line 340-341, “Using smartpca, we projected PMBB participants

on to 1000G PC space to derive eigenvectors” sounds inaccurate. Eigenvectors in PCA are PC vectors and, as the author indicated, it is the projection of a sample data on those eigenvectors obtained with 1000G data, instead of the derivation of eigenvectors.

**Have all data underlying the figures and results presented in the manuscript been provided?**

Reviewer #1: **No: ** The authors state that their data cannot be made publicly available as they contain sensitive information on human subjects, which is reasonable. The authors also state that requests from accredited researchers for access to individual-level data relevant to this manuscript can be made by contacting biobank@upenn.edu and have committed to making the code publicly available upon publication.

Reviewer #2: **No: **

PLOS authors have the option to publish the peer review history of their article (what does this mean? ). If published, this will include your full peer review and any attached files.

**Do you want your identity to be public for this peer review?** For information about this choice, including consent withdrawal, please see our Privacy Policy .

Reviewer #1: No

Reviewer #2: No

**Figure resubmission:**
---

## [Editor Report · Decision Letter 3]

Dear Dr Damrauer,

We are pleased to inform you that your manuscript entitled "Titin-Truncating variants Predispose to Dilated Cardiomyopathy in Populations Genetically Similar to African and European Reference Populations" has been editorially accepted for publication in PLOS Genetics. Congratulations!

Yours sincerely,

Todd L. Edwards

Guest Editor

PLOS Genetics

Xiaofeng Zhu

Section Editor

PLOS Genetics

Aimée Dudley

Editor-in-Chief

PLOS Genetics

Anne Goriely

Editor-in-Chief

PLOS Genetics

Comments from the reviewers (if applicable):

**Data Deposition**

http://datadryad.org/submit?journalID=pgenetics&manu=PGENETICS-D-24-00140R3

**Press Queries**

---

## [Editor Report · Acceptance letter]

PGENETICS-D-24-00140R3

Titin-Truncating variants Predispose to Dilated Cardiomyopathy in Populations Genetically Similar to African and European Reference Populations

Dear Dr Damrauer,

We are pleased to inform you that your manuscript entitled "Titin-Truncating variants Predispose to Dilated Cardiomyopathy in Populations Genetically Similar to African and European Reference Populations" has been formally accepted for publication in PLOS Genetics! Your manuscript is now with our production department and you will be notified of the publication date in due course.

With kind regards,

Olena Szabo

PLOS Genetics

On behalf of:
